# Transcriptome differentiation in *Cryptomeria japonica* trees with different origins growing in the north and south of Japan

Tokuko Ujino-Ihara[1]*, Kentaro Uchiyama[1], Seiichi Kanetani[2], Yoshihisa Suyama[3], Yoshihiko Tsumura[4]

**1** Department of Forest Molecular Genetics and Biotechnology, Forestry and Forest Products Research Institute, Tsukuba, Ibaraki, Japan, **2** Kyushu Research Center, Forestry and Forest Products Research Institute, Kurokami, Chuo, Kumamoto, Japan, **3** Field Science Center, Graduate School of Agricultural Science, Tohoku University, Osaki, Japan, **4** Faculty of Life & Environmental Sciences, University of Tsukuba, Tsukuba, Japan

* ihara_tokuko720@ffpri.go.jp

## Abstract

*Cryptomeria japonica* is a coniferous species widely distributed throughout Japan and is therefore adapted to a variety of environments. To identify genes involved in its local adaptation, individuals of different origins growing in three common gardens located in the southern, central, and northern regions of Japan were subjected to transcriptome analysis. A transcriptome assembly guided by the whole-genome sequence of *C. japonica* yielded 77,212 transcripts derived from 56,203 genes. Based on single nucleotide polymorphisms (SNPs) detected in the transcriptome data, individuals were grouped into three genetic clusters. A total of 151 SNPs associated with population differentiation were detected using pcadapt. Of these, the allele frequencies of 40 SNPs showed associations with climatic variables, and the expression levels of genes containing 9 of these SNPs were also correlated with climatic variables. To further explore transcriptomic patterns underlying adaptation, weighted gene co-expression network analysis identified 25 gene modules. A comparison between representative expression patterns of each gene module and the genetic differentiation predicted by SNPs revealed that one module exhibited a negative correlation and another a positive correlation across all three common gardens. While defense response genes were highly expressed in individuals from the Pacific Ocean side of Japan (omote-sugi), terpenoid metabolism genes were more highly expressed in individuals from the Sea of Japan side (ura-sugi). These findings suggest that local adaptation in *C. japonica* involves not only responses to abiotic stress but also a significant contribution from genes involved in responses to biotic stress.

---

---

**Data availability statement:** Raw data are available from DDBJ DRA (Accession number:DRR668422-DRR668493, DRR196892- DRR196915).

**Funding:** This study was funded by the Environment Research and Technology Development Fund (JPMEERF20S11808) of the Environmental Restoration and Conservation Agency of Japan (T.U.I. and K.U.) as well as JSPS KAKENHI Grants JP24H00527 (U.K.). The funders had no role in study design, data collection and analysis, decision to publish, or preparation of the manuscript.

**Competing interests:** The authors have declared that no competing interests exist.

## Introduction

Being sessile and long-lived organisms, trees adapt to fluctuating environments through changes in their genomic DNA. Due to genetic differentiation, the response of individuals to their surrounding environment is not uniform, even within the same species. Understanding such intraspecific genetic diversity has become important for determining the vulnerability of tree species to predicted future climate change; therefore, extensive research has been conducted to identify genetic differentiation that evolved for the adaptation to local environments. Common garden trials, in which individuals originating from different natural distribution areas are planted at the same test site, are useful for researching this topic.

Previous studies have employed common garden experiments to investigate the phenotypic responses of tree lines, and, more recently, their associations with genetic differentiation (i.e., sequence polymorphisms of genomic DNA) have also been surveyed [1–3]. These studies revealed that even wind-pollinated species such as conifers, which are characterized by low or continuous genetic differentiation, showed clear phenotypic variation along geographic or climatic clines. Furthermore, next-generation sequencing technology has enabled field transcriptome analysis of tree species grown in common gardens. Differences in the transcriptome often lead to phenotypic differences [4]. Although RNA expression can be highly variable under field circumstances, recent advances in analytical tools can associate the variation of gene expression with the phenotypic traits of source populations or climate characteristics of their region of origin [5,6]. Traits related to environmental adaptation often involve the expression of multiple genes. Weighted correlation network analysis (WGCNA) is widely used to detect such co-expressed gene networks and their association with the phenotype [7]. WGCNA has been successfully applied to plant species to find key gene candidates involved in the modulation of phenotypes [8–10].

*Cryptomeria japonica* is a coniferous species distributed in Japan and southeastern China. In Japan, the species are broadly distributed from the Aomori Prefecture (40.7333°N) to Yakushima Island in the Kagoshima Prefecture (30.2500°N) [11]. Phenotypic divergences between the populations established on the Pacific Ocean side and those on the Sea of Japan side have been reported in needle morphology [12], terpene concentration and constituents [13,14], clonal propagation [15], and growth patterns [16]. These findings suggested the presence of two main *C. japonica* groups: omote-sugi (distributed on the Pacific Ocean side) and ura-sugi (distributed on the Sea of Japan side). DNA polymorphisms also supported this genetic differentiation and suggested the existence of two additional genetically diverged groups, one originating from the northern peripheral region and the other from Yakushima Island [17–20]. The phenotypic and genetic differentiation observed in the above-mentioned studies indicated that *C. japonica* is locally adapted, and thus the genes involved in this adaptive process have been surveyed using genomic DNA polymorphisms in transcribed regions. These previous studies have identified SNPs showing signatures of genetic differentiation in adaptive genomic regions, either as outliers or through correlations with environmental variables, thereby revealing genomic regions

under strong selection within certain linkage groups. In an analysis using 148 CAPS markers, two genes were identified as potentially involved in the genetic differentiation between omote-sugi and ura-sugi, although their functions remained unknown [21]. Subsequently, a study based on 1026 SNP markers detected 12 outlier genes significantly associated with environmental variables. Among these, one showed sequence similarity to MYB transcription factors known to regulate the flavonoid biosynthetic pathway and proanthocyanidin production, suggesting a possible role in responses to biotic stress [17]. Further analysis using 3930 SNP markers identified 208 outlier genes, 43 of which were associated with environmental variables. These included several genes potentially involved in stress responses [19]. A more recent study identified 239 SNPs associated with climatic variables, highlighting the importance of winter temperature and precipitation seasonality in environmental adaptation [20]. While such studies based on genetic polymorphisms have advanced our understanding of local adaptation, gene expression was not examined in these studies, and functional interpretations of how these genes contribute to adaptation remain limited in the existing literature. Nose et al. (2023) [22] investigated seasonal variation in gene expression across different planting sites using a single clonal genotype, but did not compare individuals with different genetic backgrounds.

In the present study, we conducted high-throughput transcriptome sequencing of current-year shoots collected from *C. japonica* individuals grown in three common gardens, on days when the temperature exceeded 30 °C. These gardens were located in northern (Miyagi Prefecture), central (Ibaraki Prefecture), and southern (Kumamoto Prefecture) Japan. The genetic differentiation among individuals was evaluated based on SNPs identified by mapping RNA-Seq reads to the reference genome [23]. To investigate genes potentially involved in local adaptation, we analyzed the transcriptomic data using two complementary approaches. First, we employed a SNP-based strategy to detect outlier loci and assess associations between their genotype and climatic variables. Second, we applied weighted gene co-expression network analysis (WGCNA) to identify co-expressed gene modules and evaluated how their representative expression profiles relate to genetic differentiation and climatic variables. If local environmental selection pressures contribute to phenotypic divergence, differences in the expression of specific genes may underlie such divergence. In that case, the expression of adaptive genes would be expected to correlate with either the genetic distance among individuals or the climatic differences among their population origins. Through this integrated approach, our study identifies candidate genes associated with local adaptation, providing valuable insights for the conservation and future management of *C. japonica* genetic resources.

## Materials and methods

### Plant materials

The details related to the collection of individuals from natural populations and the establishment of a nursery are reported previously [21]. Cuttings from the nursery were grown in a field at the Forestry and Forest Products Research Institute (Tsukuba, Ibaraki, Japan) for 2 years. They were then transferred to three common gardens, one in the Ibaraki (IBR) Prefecture, one in the Kumamoto (KMT) Prefecture, and the other in the Miyagi (MYG) Prefecture, which were established based on a randomized design. Approximately 1000 trees were transplanted in each garden in 2015, 2017 and 2014, respectively. Out of the 23 populations grown in these gardens, 12 populations were selected to represent the natural distribution of *C. japonica* (Fig 1, Table 1). The map was generated using R package "maps" [24], with the map of Japan was sourced from the R package "mapdata" [25]. Two clones per population (24 clones in total) from each common garden were subjected to transcriptome analysis. Ramets from the same individual were used in all three common gardens when possible.

### Estimation of climate differences

The historical bioclimatic variables for the original locations of populations and the common gardens were extracted from WorldClim [26] at a scale of 30 arc seconds using the R package "raster" [27] (S1 Table A). Correlations among climatic

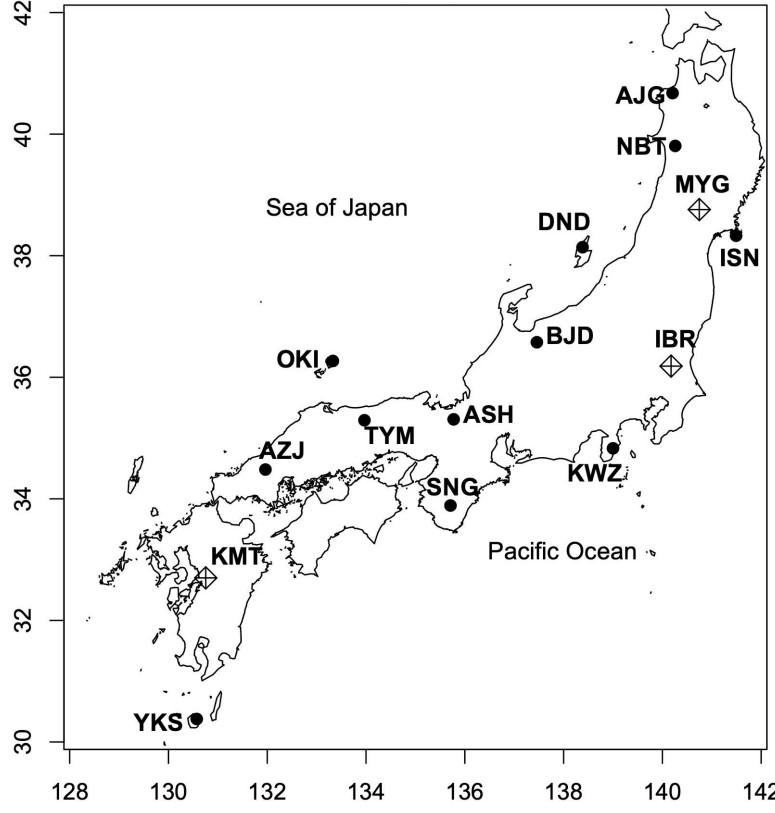

**Fig 1. Locations of source populations and common gardens.** Source populations are indicated by circles, and common gardens by diamond symbols. The map was generated using the R package "maps" [24], with the map of Japan sourced from the R package "mapdata" [25]. Both packages use public domain geographic data and are free of copyright restrictions.

variables were calculated using the cor() function in R with the "spearman" method [28] and visualized using the R package "corrplot" [29]. In addition, the bioclimatic variables of the common gardens during the period in which the material trees were grown were calculated using climatic values obtained from Agrometeorological Grid Square Data (AMGSD) [30], using the R package "dismo" [31]. It demonstrated a clear increase in temperature at each site, with an annual average temperature increase of approximately 1°C at each location (S1 Table A). The differences in climate variables between original locations and common gardens were evaluated via principal component analysis (PCA) using dudi.pca in the R package "ade4" [32] (S1 Fig). More climate transfer differences (CTD) would be more stressful conditions to the material trees and thus may shape the gene expression. To assess this hypothesis, the CTD between the original locations and the climate of the common gardens during the trial period was calculated as described in a previous study [2], using the top four principal components (PCs) from the PCA.

## RNA-Seq

Current-year shoots of well-sunned branches were sampled on June 28th, 2022, at IBR, August 8th, 2020, at KMT, and on August 2nd, 2021, at MYG. The samples were collected in these hot summer days so that the transcriptome analysis would be able to provide more information on genes involved in the response to high temperatures. Summer samples were selected for analysis because, in addition to high-temperature responses, previous studies have demonstrated regional variation in terpene biosynthesis, which is active in summer [33], as well as in growth patterns, which also tend

**Table 1. Locations of the analyzed populations and common gardens.**

| Source Populations | Abbrevi-ation | Genetic groups[a] | LAT | LON | Sampled clones | | | Number of clones |
|---|---|---|---|---|---|---|---|---|
| | | | | | IBR | KMT | MYG | |
| *Origin of populaitons* | | | | | | | | |
| Ajigasawa | AJG | ura | 40.6756 | 140.2053 | AJG033, AJG034 | AJG031, AJG034 | AJG031, AJG034 | 3 |
| Nibetsu | NBT | ura | 39.8061 | 140.2600 | NBT016, NBT028 | NBT016, NBT025 | NBT016, NBT025 | 3 |
| Ishinomaki | ISN | omote | 38.3286 | 141.4919 | ISN017, ISN019 | ISN016, ISN017 | ISN016, ISN017 | 3 |
| Donden | DND | ura | 38.1397 | 138.3833 | DND561, DND571 | DND561, DND571 | DND561, DND571 | 2 |
| Bijodaira | BJD | ura | 36.5761 | 137.4589 | BJD020, BJD025 | BJD025, BJD035 | BJD020, BJD025 | 3 |
| Kawazu | KWZ | omote | 34.8314 | 139.0000 | KWZ008, KWZ009 | KWZ008, KWZ009 | KWZ008, KWZ009 | 2 |
| Ashu | ASH | ura | 35.3078 | 135.7739 | ASH016, ASH033 | ASH016, ASH033 | ASH016, ASH033 | 2 |
| Shingu | SNG | omote | 33.8900 | 135.7100 | SNG002, SNG011 | SNG001, SNG011 | SNG001, SNG011 | 3 |
| Tsuyama | TYM | ura | 35.2931 | 133.9678 | TYM008, TYM023 | TYM019, TYM023 | TYM001, TYM005 | 5 |
| Oki | OKI | ura | 36.2683 | 133.3292 | OKI001, OKI010 | OKI001, OKI010 | OKI001, OKI010 | 2 |
| Azouji | AZJ | ura | 34.4820 | 131.9634 | AZJ006, AZJ040 | AZJ006, AZJ040 | AZJ006, AZJ040 | 2 |
| Yakushima | YKS | yaku | 30.3781 | 130.5731 | YKS002, YKS007 | YKS005[b] | YKS025, YKS032 | 5 |
| *Common gardens* | | | | | | | | |
| Ibaraki | IBR | | 36.1833 | 140.1760 | | | | |
| Kumamoto | KMT | | 32.6995 | 130.7549 | | | | |
| Miyagi | MYG | | 38.7587 | 140.7470 | | | | |

a. Genetic groups based on DNA polymorphisms.

b. Shoots from two clonal ramets of YKS005 were used in the analysis.

to intensify during this season [22]. Sampling was conducted at around 2 pm at all sites to reduce the difference caused by circadian oscillation. The air temperature at this time at IBR, KMT, and MYG was 36°C, 34°C and 30°C, respectively. Meteorological data for the 30 days preceding the sampling dates were retrieved from AMGSD [30] and visualized using base plotting functions in R [28] (S2 Fig). The most recent day of rainfall before sampling was 9 days prior in IBR, 6 days prior in KMT and 2 days prior in MYG. The humidity on the sampling day was therefore higher at MYG (S2 Table). Total RNA was extracted from the shoots using a Maxwell RSC Plant RNA Kit (AS1500) and Maxwell RSC48 instrument (Promega). In addition to being subjected to the protocols required by the kit, the samples were pre-treated with CTAB solution (2% CTAB, 2% PVP, 100 mM Tris-HCl [pH 8.0], 25 mM EDTA, 2 M NaCl, and 2% β-mercaptoethanol). Pair-ended 150-bp reads were obtained using an Illumina NovaSeq 6000 platform (Novogene). Raw data are deposited on DDBJ DRA (Accession number: DRR668422-DRR668493).

## Constructing a reference transcript set and obtaining expression data

The obtained reads were filtered using Trimgalore v 0.6.7 [34] to remove low-quality reads. Reads less than 50 nt in length were then removed using the SolexaQA++v3.1.7.1 LengthSort command [35]. A transcript sequence set was constructed from reads data obtained in this study and reads used in the previous study to improve the quality of assemblage (Accession number: DRR196892- DRR196915) [36]. The whole-genome assembly (SUGI ver.1) [23] was used as guidance for constructing transcript assemblage. First, the reads of each sample were mapped to the *C. japonica* reference genome sequence using hisat2 version 2.2.1 with default parameters [37]. Transcripts were predicted using the resultant bam files by psiclass with default parameters [38]. The gtf file generated by psiclass was then merged with standard predicted genes (SUGI_1.std.gene.gff3) [23] by stringtie version 2.2.1 with default parameters [39]. The counts for each transcript were calculated based on the merged gtf file. Sequence of predicted transcripts were extracted by gffread [39,40]. Newly predicted genes, which were not included in the SUGI_1.std.gene.gff3, were prefixed with "CJHT".

## Annotation of reference transcripts

The completeness of the assembly was assessed by comparing it with the embryophyta_odb10 database including the core gene set of land plants in BUSCO v5 with default parameters [41]. The reference sequences were compared to Araport11_pep_20220505.fa, which was retrieved from the TAIR database (www.arabidopsis.org) [42], and the UniProtKB/Swiss-Prot database [43] using blastx 2.10.0+ [44] with 1e-5 as the cutoff value.

## Single nucleotide polymorphism detection

After filtering any potential PCR duplicates using the rmdup command of samtools v1.13 [45], single nucleotide polymorphisms (SNPs) between samples were extracted via the mpileup command of bcftools v1.16 [45]. The obtained SNPs were further filtered using the filter command of bcftools with options -g 3, -G 10, and -e 'QUAL≤30||DP<100||MAF≤0.05' to obtain reliable SNPs. Discordant genotypes between ramet samples from the same individual were excluded using an in-house perl script. SNP frequency per 1kb was calculated for each polymorphic gene. SNPs from the genes with high SNP frequencies (≥10 SNPs per 1kb) were excluded to avoid possible false SNPs caused by paralogous genes or multi-gene families.

## Genetic differentiation among samples

To assess genetic differentiation among the samples examined in this study, principal component analysis (PCA) was performed on SNP data using the pca function implemented in the R package "pcadapt" [46]. To reduce redundancy caused by linkage disequilibrium (LD), SNPs with a pairwise squared correlation coefficient ($r^2$) ≥ 0.8 within a 100-kb window were identified using PLINK v1.9 [47], with the options –r2, –ld-window-kb 50, and –ld-window-r2 0.8. The resulting possible LD pairs were then represented as an undirected graph, where nodes corresponded to SNPs and edges represented LD between SNP pairs. The graph was constructed and analyzed using the igraph package in R [48]. For each connected component (i.e., a set of SNPs in strong LD), only the first SNP in genomic order was retained as a representative marker. This approach enabled the systematic pruning of SNPs in high LD, ensuring that the retained SNP set better reflected independent genetic signals. A total of 16,620 SNPs remained after this filtering step and were used for downstream analyses. The PCA results were visualized using the fviz_pca_ind function in the R package "ggplot2" [49] and "ggrepel" [50].

## Detection outlier SNPs associated with climate differences

Using the 16,620 SNPs selected as described above, we conducted outlier detection using the R package pcadapt [46] to identify loci potentially associated with local adaptation. Given the small number of individuals per population in our dataset, we chose pcadapt because it is an individual-based method that does not require predefined population grouping. Principal component analysis (PCA) based on the SNP data indicated that PC1 and PC2 captured the major genetic variation among the three genetic groups of *C. japonica.* Accordingly, the number of principal components was set to K = 2. SNPs with a false discovery rate (FDR) < 0.001 were considered statistically significant outliers. To assess the relationship between climatic variables and genotype distributions of outlier SNPs, we applied a non-parametric Kruskal–Wallis test using the kruskal.test function in R [28]. This test evaluated whether climatic values differed significantly among genotypes for each SNP. Although multiple testing correction was applied, no SNPs remained significant at FDR < 0.1. Therefore, we adopted an unadjusted p-value threshold of ≤ 0.001 to identify SNPs potentially associated with climatic variables. Because most outlier SNPs were not included in the WGCNA analysis described below, we also assessed the correlation between climatic variables and the expression levels of genes containing outlier SNPs using the corr.test function in the R package "psych", with the "spearman" method [51]. After correction for multiple testing, no SNPs remained significant at FDR < 0.1. Thus, we again used an unadjusted p-value threshold of ≤ 0.001 to identify potentially climate-associated SNPs.

To determine whether the candidate adaptive SNPs identified in this study overlapped with those reported in previous studies [17,19–21], we compared their genomic locations. Since the genomic positions of SNPs in earlier studies were not always available, we performed BLASTN searches against the reference genome using gene sequences containing significant SNPs from those studies. Additionally, because some of the candidate SNPs reported by Uchiyama et al. [20] were located in non-genic regions, we compared the genes located within 10 kb of those SNPs to the candidate genes identified in our study.

## Overall transcriptome comparison

The correlations of all expression profiles between samples were calculated using the cor function in R with the "spearman" method and visualized using the R package "corrplot" [29]. To normalize the raw count data, the variance stabilization transformation (vst) was applied via the iDEP website tool [52]. PCA was carried out for the top 1000 transcripts with the highest quartile coefficient of dispersion (QCD) between samples for each common garden using dudi.pca in the R package "ade4" [32].

## Gene expression differentiation detection by WGCNA

To reduce the effect of random fluctuation in low-expression transcripts, transcripts whose expression was lower than median expression value were first removed. We then focused on top 5000 transcripts with the highest QCD among filtered transcripts for each common garden. To construct consensus gene networks of three common gardens, the 5000 transcripts from each site were concatenated and the set consisting of 8,255 transcripts was used in WGCNA. The vst counts were used as the expression value of each transcript. WGCNA was conducted using the R package "WGCNA" to survey gene networks that showed expression patterns associated with genetic differentiation [7]. The soft-threshold power for detecting co-expressed networks (gene modules) was set to 6 in constructing a consensus network, and the networks were constructed using the blockwiseConsensusModules function with default settings, except for corType = "bicor," networkType = "signed," maxBlockSize = 10000, and maxPOutliers = 0.1. The correlation coefficients between the module eigengene (ME) of each detected gene module and the first, second PCs of the PCA using SNPs (PC1$_{SNP}$ and PC2$_{SNP}$) were calculated using the corr.test function in the R package "psych" with the "spearman" method [51]. The ME is indicative of the characteristics of each module, which gives the most representative gene expression in a module. We selected modules whose MEs showed a correlation with PCs with an qvalue of ≤ 0.1 as modules significantly associated with PCs. The correlation between climatic variables (bio1-bio19, CTD) and the eigengene expression was also calculated in the same way. The relationship between specific MEs and PC1$_{SNP,}$ and bioclimatic variables shown in Supplemental figures were plotted using ggplot2 [49] with ggrepel [50].

Candidate hub genes were selected if their module membership (MM) to the corresponding module was ≥ 0.6 and the absolute value of their gene significance (GS) was also ≥ 0.6. The genomic coordinates of the TPS03 transcripts, one of which was detected as a hub gene in the darkred module, were obtained from the GTF file generated by StringTie2, as described above. Sequence alignment of the two major transcripts was performed using the R package "msa" [53].

## Gene ontology enrichment analysis

Gene Ontology (GO) enrichment analyses were conducted using the R package "clusterProfiler" [54] for (i) genes with outlier SNPs detected by pcadapt, and (ii) genes in modules associated with PC1$_{SNP}$ and PC2$_{SNP}$. The reference transcriptome used in this study was annotated with GO terms based on the best matches to Arabidopsis reference genes, using the "GOTERM_BP_DIRECT" database of the DAVID web tool [55]. Annotation lists of all 56,203 genes and of the 6204 genes used in the WGCNA analysis were employed as background gene sets for the pcadapt and WGCNA analyses, respectively. To account for potential isoform-specific functions and the possibility that gene boundaries may be

inaccurately defined in regions containing multicopy genes, GO enrichment analysis was also performed at the transcript level for modules associated with $PC1_{SNP}$ and $PC2_{SNP.}$.

## Results

### Constructing reference sequence*s*

The average mapping rates of samples to the reference genome sequence (SUGI v1) were 95.7%, 96.6%, and 96.3% for IBR, KMT and MYG, respectively. The assembly of RNA-Seq reads guided by the reference whole-genome sequence resulted in 77,212 transcripts from 56,203 genes. The $N_{50}$ value of the merged reference transcript set was 1,962 bp. The homology search against *Arabidopsis* reference sequences and the UniProtKB/Swiss-Prot database showed that 42,150 genes (75.0%) exhibited homologous sequences in at least one database with a significant threshold (e-value < 1e-5). BUSCO analysis showed 93.3% of the 1614 conserved genes in embryophyta_obd10 were present in the constructed transcript set. Specifically, 87.4% were single-copy genes, and 5.9% were duplicate genes.

### SNP detection

A total of 110,869 polymorphic sites were detected in 13,344 genes after filtering out the sites that showed discrepant genotypes between ramets. The 1,983 insertions or deletions found in 1,701 genes were also excluded. The frequency of SNPs in a polymorphic gene ranged from 0.04 to 51.3 per 1 kb. As mentioned in the Materials and Methods section, the SNPs in genes with high SNP frequencies were excluded from further analysis to avoid possible false SNPs due to the misalignment of reads to regions with high sequence similarity, such as paralogs and multi-copy gene families. As a result, 94,732 SNPs in 12,389 genes were retained. The average SNP frequency was one SNP for every 506.4 bp.

The samples were classified into three groups by PCA using the genotypes of 16,622 representative SNPs selected as described in the Materials and Methods section (S3 Fig). The results were consistent with the previously reported classification of genetic groups: ura-sugi, omote-sugi, and individuals originating from Yakushima Island (hereafter referred to as yaku-sugi). Based on the PCA results, this study classified individuals from the northern peripheral region (AJG and NBT) as part of the ura-sugi group. The first axis ($PC1_{SNP}$) separated the three groups and showed yaku-sugi as more genetically differentiated from the other groups, whereas the second axis ($PC2_{SNP}$) separated omote-sugi from the other groups.

### Outlier SNPs reflecting genetic structure and environmental associations

The pcadapt analysis identified 151 significant outlier SNPs, which were located in 146 genes (S3 Table). Of these, 140 SNPs (in 137 genes) were associated with $PC1_{SNP}$, and 11 SNPs (in 9 genes) were associated with $PC2_{SNP}$. Gene Ontology (GO) enrichment analysis was conducted to infer the putative functions of the genes containing these SNPs; however, no GO terms were significantly enriched. Among the SNPs associated with $PC1_{SNP}$, four, five, and three were found in genes annotated with GO terms related to abiotic stress, biotic stress, and both, respectively. For the SNPs associated with $PC2_{SNP}$, two were related to abiotic stress and one to biotic stress.

The most significant SNPs associated with $PC1_{SNP}$ was located on linkage group 7 (LG7) and found within a homolog of the vascular plant one-zinc finger protein (*VOZ1*), a transcription factor known to be involved in diverse biotic and abiotic stress responses in Arabidopsis [56]. On the other hand, the most significant SNP associated with $PC2_{SNP}$ located on LG6, within a gene homologous to the Arabidopsis gene *ADAPTOR PROTEIN COMPLEX-4* (*AP-4*; AT5G11490), which has been implicated in plant immunity against avirulent bacterial pathogens [57].

The non-parametric KWtest detected possible associations between 40 of the 151 SNPs ($p \leq 0.001$) and bioclimatic variables, although none remained significant after correction for multiple testing (S3 Table). Associations were most frequently observed with temperature-related bioclimatic variables, particularly bio1 (annual mean temperature), bio4 (temperature seasonality), bio6 (min temperature of coldest month), bio7(temperature annual range), and bio11(mean

temperature of coldest quarter). It should be noted that bio1, bio6, and bio11 were highly correlated with one another, as were bio4 and bio7 (S4 Fig).

Significant correlations between expression of genes containing outlier SNPs and bioclimatic variables were detected for 9 out of the 151 SNPs ($p \leq 0.001$), although none remained significant after correction for multiple testing (S3 Table). No single gene showed a consistent correlation with bioclimatic variables across all three common gardens. Specifically, two genes showed significant correlations in IBR, four in KMT, and five in MYG, with one gene (*CJHT.1995*) overlapping between KMT and MYG. Except for one gene (*CJHT.6165*), all of these were associated with $PC1_{SNP}$ and showed higher frequencies of distinct alleles in yaku-sugi compared to the omote- and ura-sugi lineages. Seven of the nine genes were associated with temperature-related bioclimatic variables, particularly bio1 (annual mean temperature), bio6 (minimum temperature of the coldest month), and bio11 (mean temperature of the coldest quarter). One of the nine genes was related to stress response and annotated with the Gene Ontology (GO) term "cellular response to water deprivation."

Comparison with previously reported adaptive genes revealed that four of the genes containing outlier SNPs detected in this study overlapped with those identified in previous studies (S3 Table) [17,19,20]. All four SNPs were associated with $PC1_{SNP}$. Among them, only the SNP located in *CJHT.5841* was annotated with a stress-related GO term: "defense response to fungus." In addition, the genotype of one of these SNPs (*SUGI_0576340*) showed a significant association with bio12.

## Overall transcriptome profile of samples from the common gardens

After filtering low-expression transcripts, no transcripts were exclusively expressed in specific populations or genetic groups. There was a positive correlation between the overall transcriptome of the samples, with the correlation coefficients ranging from 0.75 to 0.99 (S5 Fig). Even though they were grown under different environmental conditions, ramets derived from the same individual tended to show more correlated gene expression patterns than a pair of individuals with the same origin grown in the same common garden. On the other hand, the PCA using expression data of highly variable transcripts indicated that expression patterns were associated with the genetic groups in IBR and KMT, whereas such association was not apparent in MYG (Fig 2).

## Transcripts associated with genetic differentiation

To identify gene modules exhibiting differential expression levels associated with the genetic differentiation, we examined the correlation between SNPs and the MEs of each module. A positive correlation indicated that the gene expression was higher in the omote-sugi, while a negative correlation indicates that the gene expression was higher in the ura-sugi.

A total of 25 co-expressed gene modules were detected by WGCNA, and the MEs of three, ten, and two modules were correlated with $PC1_{SNP}$ at the IBR, KMT, and MYG sites, respectively (Table 2). Among them, the ME of the grey60 module (MEgrey60) was positively correlated at all three sites, whereas MEdarkred was negatively correlated. For $PC2_{SNP}$, MEgreenyellow showed a significant correlation at IBR. No MEs showed significant correlations with $PC2_{SNP}$ at KMT or MYG.

## Genes in modules exhibiting common correlations to genetic differences at several test sites

For grey60 module, which was positively correlated $PC1_{SNP}$ in three common gardens, no Gene Ontology (GO) term was significantly enriched (S4 Table). However, gene annotation of this module revealed nine genes related to defense responses against biotic stresses, such as fungal and bacterial infections (S5 Table). In addition, three genes showed sequence homology to *CRK8* (AT4G23160), which has been implicated in disease resistance [58]. One of these *CRK8* homologs, *CJHT.4193*, was identified as a hub gene of the module in all three common gardens (S6 Table).

On the other hand, darkred module that was negatively correlated with $PC1_{SNP}$ was also no enriched with any GO term, but included various transcripts from terpenoid biosynthetic genes (S4, S7 Tables). Despite the absence of genes that met

**IBR**

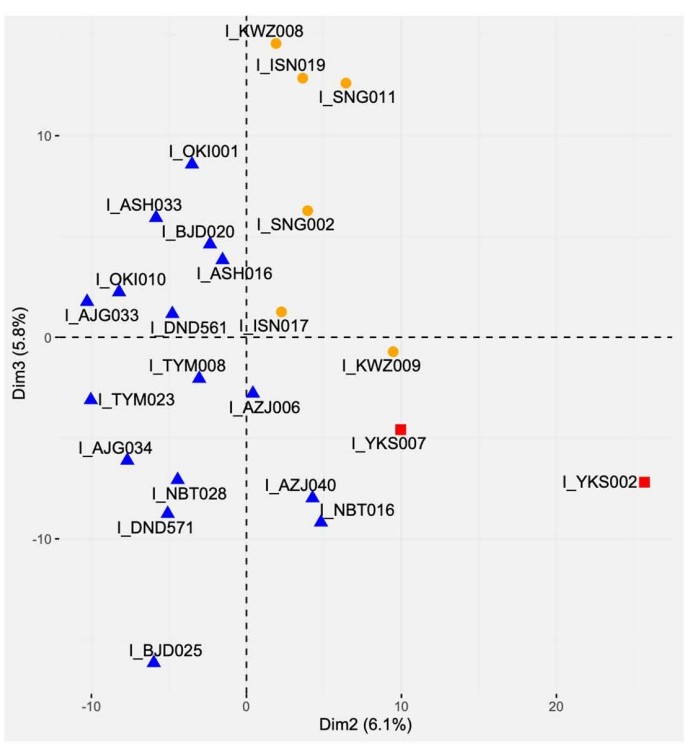

**KMT**

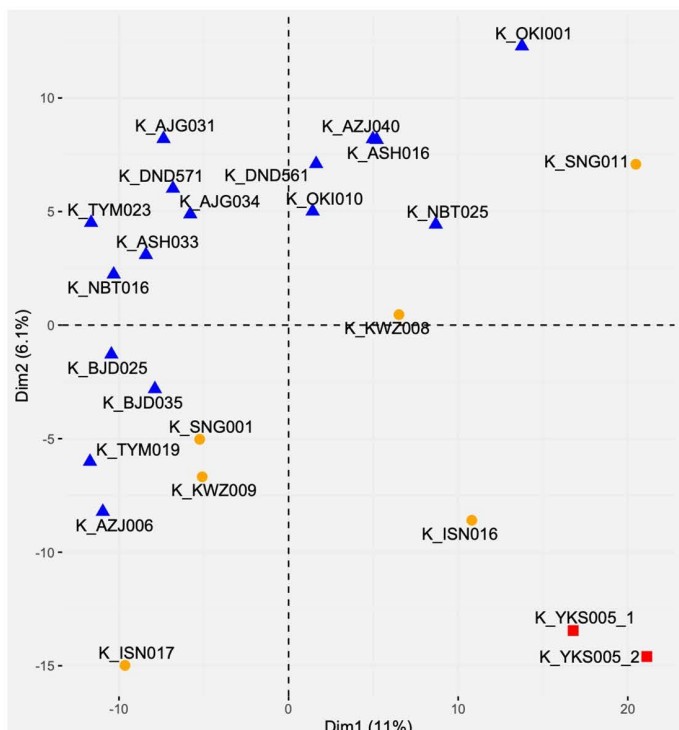

**MYG**

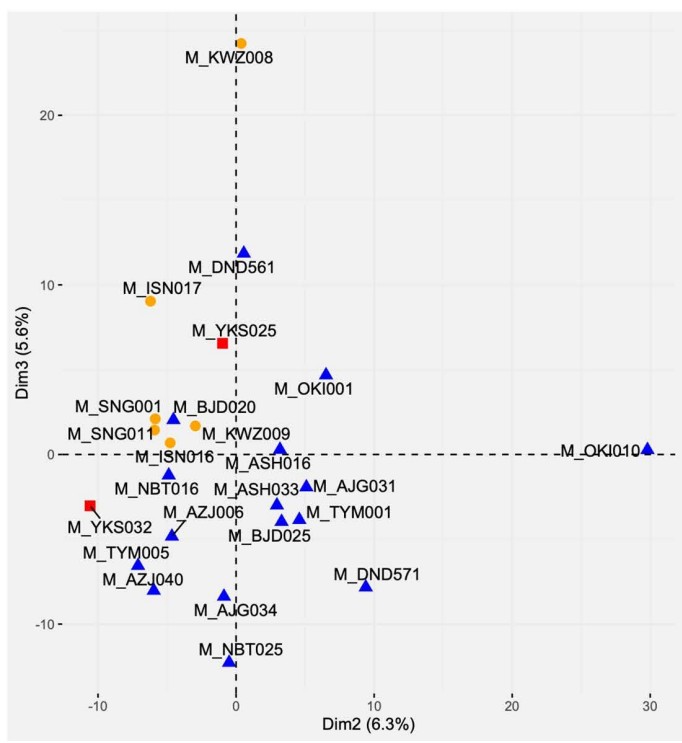

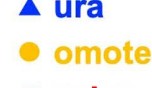

▲ **ura**
● **omote**
■ **yaku**

**Fig 2. PCA plot of samples based on transcript expression patterns.** The plot is based on the top 1,000 transcripts with the highest QCD values and sufficient expression levels (see Materials and Methods). Three genetic groups are indicated: blue triangles for ura-sugi, yellow circles for omote-sugi, and red squares for yaku-sugi.

**Table 2. Module eigengene with significant correlation to PC$_{SNP}$.**

| Modules | r | p | FDR | Transctipt numbers |
|---|---|---|---|---|
| *IBR* | | | | |
| PC1$_{SNP}$ | | | | |
| **MEgrey60** | 0.67 | 0.0003 | 0.0170 | 59 |
| **MEdarkred** | −0.66 | 0.0005 | 0.0400 | 43 |
| MEgreen | −0.57 | 0.0032 | 0.0170 | 395 |
| PC2$_{SNP}$ | | | | |
| MEgreenyellow | 0.56 | 0.0017 | 0.0044 | 152 |
| *KMT* | | | | |
| PC1$_{SNP}$ | | | | |
| **MEgrey60** | 0.78 | 0.0000 | 0.0005 | 59 |
| MEturquoise | 0.64 | 0.0008 | 0.0209 | 1646 |
| MEpurple | 0.57 | 0.0035 | 0.0506 | 153 |
| MEmagenta | 0.57 | 0.0039 | 0.0506 | 164 |
| MEtan | 0.52 | 0.0099 | 0.0857 | 133 |
| MEorange | 0.55 | 0.0050 | 0.0517 | 37 |
| MEgreenyellow | 0.50 | 0.0127 | 0.0990 | 395 |
| **MEdarkred** | −0.64 | 0.0008 | 0.0209 | 43 |
| MEred | −0.62 | 0.0012 | 0.0227 | 615 |
| MEsalmon | −0.55 | 0.0012 | 0.0517 | 87 |
| *MYG* | | | | |
| PC1$_{SNP}$ | | | | |
| **MEgrey60** | 0.72 | 0.0002 | 0.0059 | 59 |
| **MEdarkred** | −0.65 | 0.0004 | 0.0236 | 43 |
| CTD | | | | |
| MEgrey60 | 0.58 | 0.0819 | 0.0819 | 59 |

a. Module names were assigned by the WGCNA algorithm using arbitrary color labels for ease of identification.

the requisite criteria of hub-genes across all three common gardens, a gene (*CJHT.692*) with sequence similarity to Arabidopsis *terpene synthase 03* (*TPS03*, AT4G16740) demonstrated a robust correlation with the other genes and with PC1$_{SNP}$ (S6 Table). Mapping of RNA-Seq reads suggests the existence of multiple transcripts from this locus (S6 Fig). Based on the expression patterns, the main transcripts are *CJHT.692.10*, which is predicted to be the hub gene of this module in KMT and IBR, and *CJHT.692.8*. Both transcripts were expressed across all three major genetic groups. The *CJHT.692.8* was not included in the WGCNA analysis due to its relatively low QCD value.

## Heat and light response genes excessed in omote-sugi at IBR

In addition to grey60 and darkred modules, green module negatively correlated with PC1$_{SNP}$ in IBR, which enriched "signal transduction". For PC2$_{SNP}$, greenyellow module was positively correlated in IBR and the module enriched with genes responding to heat and hypoxia stress such as heat shock proteins (S4, S8 Tables). In addition to those GO terms,

greenyellow module was characterised by a high number of genes related to light response such as Chlorophyll A-B binding family proteins (S4, S8 Tables).

### More modules correlated to genetic differentiation at KMT

In KMT, an additional six modules to grey60 showed a positive correlation with PC1$_{SNP}$ (Table 2). GO enrichment analysis showed that the magenta module was enriched with defense response genes, particularly those involved in systemic acquired resistance. The tan module was associated with lipid metabolism, and the greenyellow module with heat and oxidative stress responses. No GO terms were enriched in the turquoise module; however, it contained many chloroplast-localized genes (S9 Table) and transcripts from photosynthesis-related genes (S4, S10 Tables). On the other hand, MEsalmon and MEred were negaitvely correlated with PC1$_{SNP}$. Salmon module was enriched with lipid biosynthetic related genes. No GO term enriched in red module.

### Transcripts associated with climate variables

The correlation between the climatic variables (bio1–19) and the eigengene expression of modules were assessed to explore what environmental conditions might have affected gene expression. More temperature-related variables show a high correlation with the eigengene expression than precipitation-related variables (S11 Table), especially bio4 (temperature seasonality) and bio7(annual range of air temperature). Only MEdarkred correlated with precipitation related variable, bio15 (precipitation seasonality). The strongest correlations were observed between MEturquoise and bio6 (mean daily minimum air temperature of the coldest month), and between MEgrey60 and bio4, both at KMT (S7 Fig). Correlation with CTD was detected only with MEgrey60 in MYG (Table 2).

### Comparison between SNP and transcriptome possibly associated with local adaptation

We assessed whether the genes containing outlier SNPs detected by pcadapt were also included in co-expression modules whose module eigengene (ME) expression levels showed significant correlations with genetic differentiation. Two such genes were identified (*CJHT.10423* and *CJHT.15030*; S3 Table). *CJHT.10423* was assigned to the green module, which showed higher expression in ura-sugi at the IBR site, and was homologous to the Arabidopsis SNF2 domain-containing protein/ helicase domain-containing protein *PIE1* (AT3G12810). Although *PIE1* is best known for its role in the regulation of flower development, it has also been reported to be involved in the defense response [59]. *CJHT.15030* was assigned to the red module, which also exhibited higher expression in ura-sugi at the KMT site, and showed homology to a Cwf15/Cwc15 cell cycle control family protein (*EMB2769*; AT3G13200). To date, no Gene Ontology terms related to stress response have been assigned to *EMB2769*.

## Discussion

This study employed a comparative transcriptomic approach to examine genetic differentiation among *C. japonica* individuals grown in three common gardens located in the northern, central, and southern regions of Japan. Transcriptomic data provide information on both genetic polymorphisms and gene expression differences among individuals from different geographical origins. Although the sample size was relatively small (35 individuals from 12 populations), integrating these two layers of data enabled the identification of genes potentially involved in local adaptation in *C. japonica.*

By combining outlier SNP detection using pcadapt with correlation analyses between the expression of genes containing those SNPs and bioclimatic variables, we identified loci that may contribute to climatic adaptation. Although the roles of these genes in environmental adaptation cannot be inferred from their assigned GO terms, their identification as outlier SNPs and the observed associations between their expression levels and climatic variables suggest that they are strong candidates for involvement in local adaptation.

In addition to sequence variation, we analyzed differences in gene expression, which more directly influence phenotypic traits. These differences, whether genetically determined or environmentally induced, may also reflect local adaptation. Gene expression profiles were more similar among clonal ramets grown in different gardens than among individuals from the same population or genetic group (i.e., omote-sugi, ura-sugi, and yaku-sugi) grown in the same location (S5 Fig). This suggests that gene expression is strongly influenced by genetic background. At the same time, expression differences among individuals within the same group indicate substantial genetic diversity within those groups (Fig 2).

Moreover, the correlation of expression among clonal replicates across gardens suggests that epigenetic regulation had a limited effect under the differing environmental conditions (S1 Fig). However, for genes with greater expression variability, expression patterns tended to be more similar within genetic groups at the IBR and KMT sites (Fig 2), while no such pattern was observed at MYG. On the sampling date, air temperatures were higher at IBR and KMT than at MYG (S2 Fig and S2 Table), and both sites were also likely drier. These conditions may have induced differential gene expression responses among genetic groups. The fact that the greenyellow module, which is enriched in heat-responsive genes, was correlated with PC1$_{SNP}$ at KMT and with PC2$_{SNP}$ at IBR may support the interpretation.

## Climatic variables associated with population differentiation

Among the genes containing outlier SNPs detected in this study, the genotypes of 40 SNPs showed potential associations with climatic variables, and the expression levels of genes containing nine outlier SNPs were also associated with climatic variables. The majority of associations were linked to temperature-related variables rather than to precipitation. Consistently, in the WGCNA analysis, module eigengene expression levels were more frequently correlated with temperature-related bioclimatic variables. This pattern may partly reflect a sampling bias, as tissues were collected during extremely hot summer days, and the analyzed SNPs were located in genes expressed under high-temperature conditions. Nevertheless, only two heat-responsive genes showed weak associations with temperature-related variables (S3 Table).

In a previous study by Uchiyama et al. [20], bio11 (mean temperature of the coldest quarter) emerged as the most influential variable, and in the present study, associations between bio11 and outlier SNPs were also frequently observed. As noted earlier, bio11 is strongly correlated with several other bioclimatic variables. Moreover, it is important to recognize that temperature covaries with a range of abiotic and biotic environmental factors and may indirectly shape selective pressures through associated ecological gradients. Thermal conditions can influence not only *C. japonica* but also interacting organisms such as pathogens and herbivores. Thus, in some cases, temperature itself might not be the primary driver of selection, but rather act as a proxy for a broader set of environmental pressures. Further research will be needed to clarify which environmental factors are truly critical for local adaptation in *C. japonica*.

## Outlier SNPs associated with the differentiation of yaku-sugi

PC1$_{SNP}$ clearly distinguished the omote, ura, and yaku-sugi, with particularly strong differentiation observed for the yaku-sugi. Among the highly significant outlier SNPs, many showed distinct allele frequency patterns, with allele frequencies in yaku-sugi differing markedly from those in the omote-sugi and ura-sugi groups (S3 Table). The SNP most strongly associated with PC1$_{SNP}$ was located on linkage group 7 (LG7), within a gene homologous to *vascular plant one-zinc finger protein 1* (*VOZ1*), a transcription factor broadly involved in biotic and abiotic stress responses (S3 Table) [56]. Given the regulatory role of transcription factors, and prior reports that LG7 contains genomic regions specific to yaku-sugi [19], this finding may be relevant to the adaptive differentiation of the yaku-sugi. In addition, one outlier SNP was identified in gene *CJHT.18154* on LG7, which overlaps with a previously reported candidate gene, and two more outlier SNPs were located in MYB-related transcription factor genes on LG7 (*SUGI_0753470* and *SUGI_0734910*; S3 Table). While further validation is required, these findings suggest that LG7 may play an important role in the genetic basis of yaku-sugi differentiation.

 

## Biotic stress response genes involved in the differentiation of omote-sugi

The outlier SNP most strongly associated with $PC2_{SNP}$, which distinguishes omote-sugi from the other groups, was located in a gene homologous to AP-4, a gene linked to hypersensitive cell death in plant immunity [57]. This raises the possibility that selection mediated by pathogens may have contributed to the differentiation of omote-sugi.

We also found that the grey60 module, which includes many defense response genes and was correlated with $PC1_{SNP}$, showed higher expression in some omote-sugi individuals than in yaku-sugi across the common gardens (S8 Fig). The hub gene of this module is similar to the retrotransposon-like region of *CRK8* of Arabidopsis. Although the exact function of this gene has not been determined, an association between polymorphisms in the retrotransposon-like region of *CRK8* and quantitative disease resistance to the fungal pathogen *Sclerotinia sclerotiorum* was previously reported in Arabidopsis [58]. The author speculated that the retrotransposon region is not actually transcribed, but a corresponding region is likely transcribed in *C. japonica*. The region includes a Reverse transcriptase- and Ribonuclease H-like domains, and is conserved among a wide range of plant species. The sequences with similarities to *CRK8* are distributed in the *C. japonica* genome at a relatively high frequency. The role of this gene in defense response is highly intriguing.

As shown in a previous study [14], the pathogenic composition associated with *C. japonica* varies regionally. *Fomitiporia torreyae*, the causal agent of wood decay in *C. japonica*, prefers high temperatures and tends to occur in regions where winter temperatures remain above –5°C [60]. Its establishment is therefore more likely in areas currently inhabited by the omote-sugi lineage. The consistently higher expression of disease resistance–related genes in omote-sugi than in ura-sugi across all three common gardens may reflect a constitutive adaptive response to pathogens that commonly co-occur with omote-sugi, such as *F. torreyae.*

In contrast to this consistent pattern observed at all three sites, the magenta module, which contains many defense-related genes and particularly includes those involved in systemic acquired resistance, showed a moderate positive correlation with $PC1_{SNP}$ only at the KMT site. This site-specific expression pattern may result from selective infection of omote-sugi by locally occurring pathogens present at KMT.

## Ura-sugi had higher expression of terpenoid biosynthesis genes

The one of key gene of darkred module that showed a positive correlation with $PC1_{SNP}$ was involved in terpenoid biosynthesis gene. Terpenoids have a wide range of roles in responding to biotic and abiotic stress [61]. Although terpenoids have function in protection from high temperature [62], genes included in the module more related to biotic stress (S7 Table). The stored and volatile terpenoids of *C. japonica* were examined in the summer of 2019 for 12 populations at MYG, including all those analyzed in the present study except for BJD and TYM [14]. In that study, the amount of stored terpenoids was negatively correlated with warm and less snowy climates. It is consistent with the results obtained in this study, but the volatile terpenoids showed different geographic patterns.

Arabidopsis *TPS03*, which is homologous to the candidate hub gene in the darkred module, is involved in ecotypic variation in herbivore-induced volatile terpene emissions. Although differences in volatile terpene levels may reflect the amount of stored terpenes, the functional significance of the observed gene expression differences in *C. japonica* remains uncertain. In Arabidopsis, variation in terpene emission has been attributed to allelic differences in the closely related and tandemly duplicated terpene synthase genes *TPS02* and *TPS03* [63]. In *C. japonica,* transcriptional patterns in the corresponding region are also complex. The two major transcript variants do not share overlapping exons and encode distinct amino acid sequences (S6 Fig). These patterns suggest that multiple functional isoforms may exist. To clarify the structure and diversity of these transcripts, future studies should include long-read RNA sequencing such as Iso-Seq, along with resequencing of the genomic region in multiple individuals.

Regional variation in biotic stress may influence gene expression, but the observed differences in terpene biosynthesis–related genes may also be explained by the growth differentiation balance hypothesis [64]. This hypothesis proposes that plants with slower growth tend to allocate more resources to secondary metabolism. Ura-sugi is known to grow more slowly than omote-sugi during early development [16], and growth measurements at the KMT and MYG sites, though limited to early stages, support this trend. Ura-sugi may therefore produce higher levels of secondary metabolites such as terpenoids as a constitutive defense strategy.

The darkred module includes a homolog of *HOG1* (*CJHT.13749*), a gene that interacts with cytokinins and whose overexpression in Arabidopsis has been associated with reduced biomass and fewer leaves [65]. It also contains an *ADR1-L2* homolog (*CJHT.13261*), a gene known to play a central role in plant immune responses [66]. Since the module includes genes involved in both defense signaling and growth regulation, it is not possible to determine from the present results whether the higher expression of terpenoid biosynthesis genes observed in ura-sugi reflects an adaptation to specific biotic stressors, or rather a constitutive increase in defense-related compounds associated with inherently slower growth. To better understand the functional relevance of this module, future studies should investigate developmental and seasonal variation in gene expression in both omote-sugi and ura-sugi. It will also be important to examine differences in biotic stress at the origin sites of each lineage, and to assess their responses to controlled environmental stress conditions.

### Differentially expressed abiotic stress response genes between genetic cluster

As noted above, there was evidence of constitutive differentiation in the gene expression associated with biotic stresses between genetic groups, but differences in gene expression in response to abiotic stresses were also observed at KMT. At the KMT site, turquoise module, enriched with transcripts respond to reactive oxygen species and photosynthesis, tend to express higher in yaku- and omote-sugi (Table 2, Table S4B, S7 Fig). The plants adjusted their photosynthetic characteristics to their growth temperatures [67]. There were high correlations between the MEturquoise and the bio-climatic variables related to the temperature (S11 Table, r>=0.7 for bio4, bio6, bio7, and bio11, and r>=0.6 for bio1). The photosynthetic process genes of individuals from warmer regions may be adapted to high temperatures, whereas individuals from cooler regions may lack such adaptations. These differences in temperature adaptation are likely to contribute to the observed variation in gene expression at KMT. Contrary to our expectation, however, there was no correlation with bio5, which measures temperature in the warmest season (summer). Given the relatively limited variation in bio5 values across different regions (S1 Table), it is plausible that the regional adaptation of photosynthesis might be influenced by winter temperatures.

### Putative adaptive SNPs and genes in modules associated with genetic difference

We employed two complementary approaches to identify genes involved in environmental adaptation in *C. japonica.* These included the detection of SNPs potentially contributing to population genetic differentiation, and the identification of co-expressed gene modules associated with this differentiation. Genes identified by both approaches are more likely to play a significant role in environmental adaptation. However, comparison of the results revealed only two overlapping genes (*CJHT.10423* and *CJHT.15030*). Even when all SNPs in strong linkage disequilibrium (r² ≥ 0.8) with identified outlier SNPs were also considered, the result remained unchanged. Of these, one has been implicated in biotic stress responses, while the function of the other remains unclear.

One reason for the limited overlap between the two approaches is that, in the WGCNA analysis, genes with low expression levels were filtered out in the initial step, and the analysis was restricted to approximately 8,255 genes with high expression variability. In fact, among the 151 genes harboring SNPs detected by pcadapt, only 10 were included in the WGCNA gene set. Furthermore, because the SNPs used in this study are located within transcribed regions, the limited overlap between pcadapt and WGCNA may be attributed to the fact that regulatory regions controlling gene expression are

typically located outside of coding sequences. Conifers generally possess long introns and exhibit low levels of linkage disequilibrium [68], which may also contribute to the limited concordance observed. To further investigate the lineage-specific differences in gene expression revealed in this study, it will be necessary to analyze SNPs located in regulatory regions.

## Conclusion

It should be noted that the results presented here are based on limited data: transcriptome analysis was conducted only once at each common garden, and the number of individuals analyzed was relatively small. To better understand environmental adaptation in *C. japonica*, it will be necessary to obtain data from a larger number of individuals and across multiple time points. Further validation of gene expression using methods other than RNA-Seq, as well as investigation into how the outlier SNPs are linked to functional changes in gene activity, will be essential. Nevertheless, this study demonstrated the utility of field-based transcriptome analysis for identifying candidate adaptive genes. Our findings suggest that genes related not only to abiotic stress but also to biotic stress may be differentiated among lineages. Compared to *C. japonica*, which has a long generation time, pathogens and insects with much shorter life cycles may be able to respond more rapidly to ongoing environmental changes, potentially leading to increases in their populations and shifts in host–pathogen interactions. For example, *F. torreyae,* which is currently constrained by low winter temperatures [60], may expand into regions that are presently too cold for its survival as global warming progresses. *C. japonica* lineages currently distributed in these regions may lack adaptive genetic elements conferring resistance to *F. torreyae*, and could therefore suffer serious damage if such pathogens expand their range.

These results highlight the need to expand research beyond abiotic stress responses and to deepen our understanding of lineage-specific responses to biotic stresses. Incorporating the role of biotic stress in shaping local adaptation will be critical for predicting the future vulnerability of *C. japonica* populations. Therefore, adaptation to biotic stresses, in addition to abiotic stresses, must be a central consideration in future conservation and management strategies for this species.

## Supporting information

**S1 Fig. Climatic differences between the locations of source populations and common gardens.** PCA was conducted using 19 climatic variables obtained from WorldClim [26] or calculated from current climate data retrieved from AMGSD (denoted by _C) [30]. Source populations are color-coded by genetic group: blue (ura-sugi), yellow (omote-sugi), and red (yaku-sugi). Common garden sites are shown in dark green (growth period) and light green (past climate).
(PPTX)

**S2 Fig. Weather conditions for 30 days before sampling.** A. max air temperature of the day, B total precipitation of the day. Weather data for each common garden obtained from AMGSD [30].
(PPTX)

**S3 Fig. Principal components analysis using unlinked SNPs ($r^2 < 0.8$). S**cores of each individual for PC1 and PC2 were calculated using pcadapt [46]. Three genetic groups were shown in blue (ura-sugi), yellow (omote-sugi), and red (yaku-sugi).
(PPTX)

**S4 Fig. Correlations between bioclimatic variables.** Darker green indicates higher positive correlations.
(PPTX)

**S5 Fig  Correlation of overall expression profiles between samples.** Correlation was calculated based on variance-stabilized transformation (VST) counts. Darker green indicates a higher correlation.
(PPTX)

**S6 Fig.  Comparison of *TPS03* isoforms. A**. Predicted exon structures of *C. japonica TPS03* transcripts, **B**. alignment of amino acid sequences of two major transcripts, CJHT.692.8 and CJHT.692.10. In panel A, "Exon start" indicates the start positions of exons based on the *C. japonica* reference genome, and "Expression" represents the summed TPM values across all samples for each isoform.
(PPTX)

**S7 Fig.  Correlation between MEs and bioclimate variables. A.** MEturquoise in KMT and bio6 (mean daily minimum air temperature of the coldest month), **B.** MEgrey60 in KMT and bio4 (temperature seasonality). Three genetic groups were shown in blue (ura-sugi), yellow (omote-sugi), and red (yaku-sugi).
(PPTX)

**S8 Fig.  The relationship between MEgrey60 and PC1$_{SNP}$.** PC1 scores for each individual were calculated using pcadapt [46]. Three genetic groups were shown in blue (ura-sugi), yellow (omote-sugi), and red (yaku-sugi).
(PPTX)

**S1 Table.    A. Bioclimatic variables at the locations of source populations and common gardens, B. The description of bioclimatic variable.**
(XLSX)

**S2 Table.  The climatic data of the sampling day.**
(XLSX)

**S3 Table.  Outlier SNPs detected by pcadapt.**
(XLSX)

**S4 Table.  A. Enriched biological pathway GO terms in genes from modules associated with PC1$_{SNP}$ and PC2$_{SNP}$, B. Enriched biological pathway GO terms in transcripts from modules associated with PC1$_{SNP}$ and PC2$_{SNP}$.**
(XLSX)

**S5 Table.  Annotation of genes in grey60 module.**
(XLSX)

**S6 Table.  Hub gene candidates in modules associated with genetic differentiation.**
(XLSX)

**S7 Table.  Annotation of transcripts in darkred module.**
(XLSX)

**S8 Table.  Genes related to hypoxia, heat, and light stress response in greenyellow.**
(XLSX)

**S9 Table.  Enriched cellular component GO term in genes in turquoise module.**
(XLSX)

**S10 Table.  Genes related to reactive oxygen species response and photosynthesis, in turquoise module.**
(XLSX)

**S11 Table. Detected correlation between eigengene expression and bioclimatic variables.**
(XLSX)

## Acknowledgments

The author thanks Ms. Furusawa for her excellent assistance with the experiments. The authors are grateful to Y. Matsui, K. Yokoo, and R. Kusano from Kumamoto Prefecture Forestry Research and instruction Center, T. Sasaki and K. Sato from Kawatabi Field Center of Tohoku University for the maintenance and preparation of research materials.

## Author contributions

**Conceptualization:** Tokuko Ujino-Ihara, Yoshihiko Tsumura.

**Data curation:** Tokuko Ujino-Ihara, Kentaro Uchiyama.

**Formal analysis:** Tokuko Ujino-Ihara.

**Funding acquisition:** Tokuko Ujino-Ihara, Kentaro Uchiyama.

**Investigation:** Tokuko Ujino-Ihara, Kentaro Uchiyama.

**Project administration:** Tokuko Ujino-Ihara.

**Resources:** Kentaro Uchiyama, Seiichi Kanetani, Yoshihisa Suyama, Yoshihiko Tsumura.

**Writing – original draft:** Tokuko Ujino-Ihara.

**Writing – review & editing:** Kentaro Uchiyama, Seiichi Kanetani, Yoshihisa Suyama, Yoshihiko Tsumura.

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
