## [Decision Letter · Decision Letter 0]

4 Apr 2025

Dear Dr. Ujino-Ihara,

Thank you for submitting your manuscript to PLOS ONE. After careful consideration, we feel that it has merit but does not fully meet PLOS ONE’s publication criteria as it currently stands. Therefore, we invite you to submit a revised version of the manuscript that addresses the points raised during the review process.

We look forward to receiving your revised manuscript.

Kind regards,

Shailender Kumar Verma, Ph.D.

Academic Editor

PLOS ONE

Additional Editor Comments (if provided):

Reviewers' comments:

Reviewer's Responses to Questions

**Comments to the Author**

1. Is the manuscript technically sound, and do the data support the conclusions?

Reviewer #1: Partly

Reviewer #2: Yes

Reviewer #3: Yes

2. Has the statistical analysis been performed appropriately and rigorously?

Reviewer #1: Yes

Reviewer #2: Yes

Reviewer #3: Yes

3. Have the authors made all data underlying the findings in their manuscript fully available?

Reviewer #1: Yes

Reviewer #2: No

Reviewer #3: Yes

4. Is the manuscript presented in an intelligible fashion and written in standard English?

Reviewer #1: Yes

Reviewer #2: Yes

Reviewer #3: Yes

Reviewer #1: 1.Limited sample size (24 clones) and single-timepoint sampling may reduce the robustness of environmental response conclusions.

2.Potential batch effects (e.g., sampling in different years) are not addressed, which may confound expression patterns.

3.Lack of functional validation (e.g., CRISPR or qPCR) for hub genes weakens mechanistic claims.

4.Insufficient discussion on the evolutionary implications of cryptic northern refugia hypothesis mentioned in cited works.

Reviewer #2: The manuscript states that sequence data will be available through DDBJ, but no project identifier or accession numbers are provided, so at the time of review, the data are not yet available to the public.

The authors acknowledge that because their experiment included only 24 individuals from 12 populations, they cannot regard the associations they identify as proven, but instead these associations represent hypotheses to be tested in future experiments. Much more could be done to generate more specific and easily-tested hypotheses using the RNA-seq data described by the authors. The reported correlations (positive and negative) of module eigengene expression levels with principal components of the full set of SNPs are an interesting way of demonstrating a relationship between gene co-expression networks and SNP variation, but that analysis does not address obvious questions relevant to the overall effort to identify genes involved in local adaptation of sugi to different regions in Japan.

For example, one question that could be addressed is whether any of the SNPs identified by the authors in the predicted RNA transcripts show association with any of the climate variables analyzed. 24 samples is not a lot, so these tests of association will not be powerful, but the analysis can be done, and the transcripts in which any putatively-associated SNPs are found examined for evidence of climate-related differential expression. Putative transcripts that show evidence of climate-related differential expression, and which also harbor climate-associated SNP variation, are doubly-implicated in a potential role in climate adaptation. Identifying up to a few hundred such candidates would be excellent preparation for a follow-up experiment to use targeted sequencing to measure gene expression levels and SNP genotypes in those candidate genes across a much larger population. These authors are well-prepared for such an experiment, given their past history of publications.

Not all genes that show differential expression related to climate variable will harbor SNPs that are associated with climate variables, of course. Regulatory variation in how genes are expressed can be independent of structural variation in the products of the expressed genes. Gene Ontology enrichment analysis can be used to find shared metabolic, biological and cellular functions between the differentially-expressed transcripts and the transcripts bearing climate-associated SNPs, to address the question of whether the two different lines of evidence are both supporting the involvement of a common set of GO categories. Again, these results will not be solid proof of involvement, due to the small sample size, but they will represent discrete testable hypotheses to be explored in future work.

Reviewer #3: This work on transcriptome differentiation in Cryptomeria japonica offers important new insights on local adaptation of this conifer species to various environmental circumstances in Japan. One strength of the work is the method of employing ordinary gardens and assessing gene expression as well as genetic diversity. I have certain comments arranged by parts below:

Introduction:

*The introduction gives appropriate background on the significance of local adaptation in trees and intraspecific genetic diversity.

*In line 77, it would be strengthened by mentioning, in passing which genes or roles were found in earlier studies on environmental adaptability in C. japonica.

Materials and methods:

*Line 126's basis for gathering samples on hot summer days is scant. It would be helpful to clarify why for this work particular attention to capturing gene expression under high-temperature circumstances is especially crucial.

*Information on the parameters used in bioinformatic programs (hisat2, psiclass, stringtie) is lacking in the part on "Constructing a reference transcript set" lines 138–151. Reproducibility depends on this material.

*"Reads are under registration process" states line 143. Before publication, this should be revised including the accession numbers.

Results:

*The results' presentation is neat and unambiguous.

*For readers not familiar with WGCNA nomenclature, a footnote outlining the meaning of MEgrey60, MEdarkred, etc., would be helpful in Table 2.

*It would be helpful to provide more information on the particular genes involved, maybe in a supplementary table, especially lines 304–313 where modules linked with heat and light stress responses are discussed.

Discussion:

*The discussion is thorough and properly links the findings with the body of current knowledge.

*The section "Ura-sugi had higher expression of terpenoid biosynthesis genes" (lines 391–415) addresses two potential ideas to help to explain the noted trends. They do not, however, arrive at a definitive answer regarding which is more likely. A closer examination or recommendations for next research projects capable of answering this issue would be rather helpful.

*In line 467, "disrupting the existing adaptive relationships [53]" - more discussion on just how climate change might impact these particular adaptive relationships found in this study would be helpful.

tables and figures:

*For one to grasp the geographical distribution of the populace, figure 1 is simple and practical.

*Though the tiny indicators could be challenging to identify, Figure 2 clearly displays the variations in gene expression patterns among genetic groupings.

*As said before, Table 2 requires an explanatory comment for the module nomenclature.

Particular suggestions for strengthening the manuscript:

*Lines 126–131 in the Materials and Methods section should provide more specifics regarding the environmental conditions under sampling. While air temperature is stated, it would be helpful to include relative humidity data since this can greatly influence gene expression connected to water stress.

*Lines 277–288 in the Results section indicate that although no GO word enrichment was seen when characterising genes enriched in the grey60 module, 24 genes linked to disease resistance are subsequently discussed. In absence of GO enrichment, how were these genes shown to be disease-related? Better explanation of this seeming paradox would help.

*Lines 353–370 in the Discussion section, when addressing differential expression of disease resistance genes in omote-sugi, could be strengthened by tying these results to particular climatic or ecological data that might help to explain why this adaptation would be beneficial in areas where this genetic group predominates.

*Without further information, line 404 states, "mapping of RNA-Seq reads predicts the existence of multiple transcripts of this locus". Perhaps in a supplementary table further details regarding these few transcripts and their putative functional relevance would be beneficial.

This is a well-designed and carried out study with overall great value on the genetic basis of local adaptation in C. japonica. With the minor revisions suggested, it will be a significant contribution to the literature on forest genomics and plant adaptation.

**Do you want your identity to be public for this peer review?** For information about this choice, including consent withdrawal, please see our Privacy Policy

Reviewer #1: No

Reviewer #2: **Yes: ** Ross W. Whetten

Reviewer #3: **Yes: ** José Alejandro Ruiz-Chután

---

## [Author Response · Author response to Decision Letter 1]

22 Jul 2025

Responses to Editor comment

In addition to responding to the reviewers’ comments, we have also carefully reviewed and revised the main text and reference formatting to ensure consistency and compliance with the PLOS ONE guidelines. Furthermore, we would like to note that the map shown in Figure 1 was generated using the R packages “maps” and “mapdata,” which provide public domain geographic data that are free of copyright restrictions. This has now been clearly stated in the figure legend.

Response to reviewers

We sincerely thank the reviewers for their thorough and constructive comments. All points raised have been carefully addressed in the following response.

As part of a major revision, we added a new analysis of the transcriptome-derived SNPs, taking into account the comments from Reviewer 2. To improve the reliability of the results, we first reduced linkage among SNPs using PLINK and performed PCA using a set of SNPs with low linkage disequilibrium in the revised manuscript. Based on the updated principal components (PC1 and PC2), we re-analyzed the correlations with the eigengene expression patterns of the gene modules identified by WGCNA. While the overall trends were consistent with the previous results, some correlations were no longer statistically significant—specifically, the correlation between PC2 and the purple module in IBR, and between PC2 and the white module in both IBR and KMT. Conversely, a new significant correlation was observed between PC1 and the greenyellow module in KMT. In addition, Gene Significance values for PC1 and PC2 changed for some genes, and because we adopted a more stringent threshold (raising it from 0.5 to 0.6), the set of identified hub genes was also partially revised. These changes are indicated in the “Revised Manuscript with Track Changes,” where deleted text is shown with strikethrough and newly added text is shown in red.

Responses to reviewer 1

1. Limited sample size (24 clones) and single-timepoint sampling may reduce the robustness of environmental response conclusions.

We agree with this comment. For this reason, we emphasized at the beginning of the conclusion (lines L621–L624) that our findings are limited and that further analyses are necessary.

2. Potential batch effects (e.g., sampling in different years) are not addressed, which may confound expression patterns.

We acknowledge the validity of this concern. Indeed, gene expression differences among common gardens can result from multiple factors. However, our analysis focused primarily on differences in gene expression observed across genetic groups that showed consistent patterns across test sites, as these may reflect genetically fixed differences associated with environmental adaptation. In cases where gene function and environmental differences suggested plausible causes, we also discussed common garden-specific expression patterns. Furthermore, the high correlation in gene expression between ramets across different test sites suggests that batch effects may not have had a major influence for many genes.

3. Lack of functional validation (e.g., CRISPR or qPCR) for hub genes weakens mechanistic claims.

We fully acknowledge this point. However, before conducting validation using CRISPR or qPCR, we believe it is necessary to perform follow-up experiments with more time points and larger sample sizes. We have added a statement to the Conclusion highlighting the need for such additional experiments (lines L624–L626).

4. Insufficient discussion on the evolutionary implications of cryptic northern refugia hypothesis mentioned in cited works.

Among the populations included in our analysis, AJG and NBT have previously been considered to be strongly influenced by cryptic northern refugia. However, in both the SNP-based and gene expression-based PCA, these populations did not form distinct clusters but were instead grouped with the ura-sugi lineage. Therefore, in our analyses, we treated AJG and NBT as part of the ura-sugi group. Therefore, we think we were unable to evaluate the evolutionary implications of the cryptic northern refugia hypothesis based on gene expression patterns during the summer season.

Responses to reviewer 2

The manuscript states that sequence data will be available through DDBJ, but no project identifier or accession numbers are provided, so at the time of review, the data are not yet available to the public.

The registered accession numbers are provided (L169, L176, L659-660).

The reviewer acknowledges the limitations posed by the small sample size (24 individuals from 12 populations) and agrees that the detected associations should be considered as hypotheses rather than definitive conclusions. However, the reviewer encourages us to explore the RNA-seq data more thoroughly to generate additional, testable hypotheses. Specifically, they suggest examining whether SNPs found in the predicted transcripts are associated with climate variables, and whether those transcripts also exhibit climate-related differential expression. Transcripts that meet both criteria would be strong candidates for future targeted studies on climate adaptation. The reviewer also recommends performing Gene Ontology enrichment analysis to identify shared biological functions between differentially expressed transcripts and those bearing climate-associated SNPs. While acknowledging the statistical limitations, the reviewer emphasizes that such analyses could help refine hypotheses for future validation studies.

Thank you for your valuable and insightful comments. As the reviewer pointed out, the number of individuals analyzed per population in this study was limited. In addition, the sample sizes were not balanced across populations (we have added the information to Table 1), making it difficult to obtain reliable results using methods such as LFMM. Therefore, we employed pcadapt, an individual-based method, to detect outlier loci. We then assessed the associations between environmental variables and SNP genotypes using the non-parametric Kruskal–Wallis test and examined the correlation between climatic variables and the expression levels of genes containing outlier SNPs using Spearman’s method.

Although we were able to identify candidate genes potentially associated with climatic variables, it was difficult to determine their specific functional roles or involvement in environmental adaptation based solely on the results of this study.

We have revised the manuscript accordingly and added relevant descriptions in the Methods (L220-L244), Results (L323-L359,L438-L449), and Discussion (L453-L464, L481-L499,L501-L514,L599-L618) sections to clarify these points.

Responses to reviewer 3

Comment #1

In line 77, it would be strengthened by mentioning, in passing which genes or roles were found in earlier studies on environmental adaptability in C. japonica.

Thank you for your comment. In accordance with your suggestion, we have provided a more detailed explanation of the findings from previous studies and added references (L84-97, Ref. 17 and Ref. 22).

Comment #2

Line 126's basis for gathering samples on hot summer days is scant. It would be helpful to clarify why for this work particular attention to capturing gene expression under high-temperature circumstances is especially crucial.

In addition to enabling us to observe gene expression responses to high temperatures, which are important for conservation under global warming, we chose to sample in summer based on previous studies indicating that genes involved in terpene biosynthesis and growth—traits that differ between omote-sugi and ura-sugi—are activated during this season. We have added this explanation to the main text (L156-L159).

Comment #3

Information on the parameters used in bioinformatic programs (hisat2, psiclass, stringtie) is lacking in the part on "Constructing a reference transcript set" lines 138–151. Reproducibility depends on this material.

As the analyses using these software tools were performed with default parameters, we have added a statement to that effect in the main text (L180,L181,L182).

Comment #4

Reads are under registration process" states line 143. Before publication, this should be revised including the accession numbers.

As mentioned above, The registered accession numbers are provided (L169, L176, L659-660).

Comment #5

For readers not familiar with WGCNA nomenclature, a footnote outlining the meaning of MEgrey60, MEdarkred, etc., would be helpful in Table 2.

Thank you for your comment. We have added an explanation of the module naming convention to Table 2.

Comment #6

It would be helpful to provide more information on the particular genes involved, maybe in a supplementary table, especially lines 304–313 where modules linked with heat and light stress responses are discussed.

Information on the genes included in the grey60 and darkred modules, genes related to heat and light stress responses in the greenyellow module, and genes related to photosynthesis and reactive oxygen species response in the turquoise module has been added as S5, S7,S8 and S10 Table.

Comment #7

The section "Ura-sugi had higher expression of terpenoid biosynthesis genes" (lines 391–415) addresses two potential ideas to help to explain the noted trends. They do not, however, arrive at a definitive answer regarding which is more likely. A closer examination or recommendations for next research projects capable of answering this issue would be rather helpful.

Thank you for your valuable comment. Based on the current data, it is difficult to determine which interpretation better explains the elevated expression of the darkred module observed in ura-sugi. As you suggested, further analyses are needed. We have added a corresponding discussion to the revised manuscript�L571-L581�.

Comment #8

In line 467, "disrupting the existing adaptive relationships [53]" - more discussion on just how climate change might impact these particular adaptive relationships found in this study would be helpful.

In accordance with the reviewer’s comment, we have revised the relevant section to clarify how climate change may influence the relationship between Cryptomeria japonica and biotic stressors (L628-L635).

Comment #9

Lines 126–131 in the Materials and Methods section should provide more specifics regarding the environmental conditions under sampling. While air temperature is stated, it would be helpful to include relative humidity data since this can greatly influence gene expression connected to water stress.

We have added the information on the sampling dates in S2 Table. We also included the temperature and precipitation trends for the 30 days prior to sampling as S2 Fig.

Comment #10

Lines 277–288 in the Results section indicate that although no GO word enrichment was seen when characterising genes enriched in the grey60 module, 24 genes linked to disease resistance are subsequently discussed. In absence of GO enrichment, how were these genes shown to be disease-related? Better explanation of this seeming paradox would help.

Although the GO enrichment analysis did not identify statistically significant terms, we found that nine genes in the grey60 module were annotated with GO terms related to defense responses. Furthermore, our literature review revealed three additional genes that may be involved in disease resistance. These genes are now listed in S5 Table, and we have revised the main text accordingly to clarify this point. In addition, we have replaced the term “disease resistance” with “defense response” to maintain consistency with the annotated GO term (L392-L398).

Comment #11

Lines 353–370 in the Discussion section, when addressing differential expression of disease resistance genes in omote-sugi, could be strengthened by tying these results to particular climatic or ecological data that might help to explain why this adaptation would be beneficial in areas where this genetic group predominates.

In response to the reviewer’s suggestion, we added a discussion citing a study reporting that a C. japonica pathogen is more likely to occur in areas where the omote-sugi lineage predominates, which may help explain the observed patterns of disease resistance gene expression�L533-L538�.

Comment #12

Without further information, line 404 states, "mapping of RNA-Seq reads predicts the existence of multiple transcripts of this locus". Perhaps in a supplementary table further details regarding these few transcripts and their putative functional relevance would be beneficial.

We summarized the isoforms identified in the present transcriptome analysis in S6 Fig. We have revised the text accordingly (L276-278, L401-L407).

---

## [Decision Letter · Decision Letter 1]

26 Aug 2025

Transcriptome differentiation in Cryptomeria japonica trees with different origins growing in the north and south of Japan

PONE-D-25-09385R1

Dear Dr. Ujino-Ihara,

We’re pleased to inform you that your manuscript has been judged scientifically suitable for publication and will be formally accepted for publication once it meets all outstanding technical requirements.

Kind regards,

Shailender Kumar Verma, Ph.D.

Academic Editor

PLOS ONE

Additional Editor Comments (optional):

Reviewers' comments:

Reviewer's Responses to Questions

**Comments to the Author**

Reviewer #2: All comments have been addressed

2. Is the manuscript technically sound, and do the data support the conclusions?

Reviewer #2: Yes

3. Has the statistical analysis been performed appropriately and rigorously?

Reviewer #2: Yes

4. Have the authors made all data underlying the findings in their manuscript fully available?

Reviewer #2: Yes

5. Is the manuscript presented in an intelligible fashion and written in standard English?

Reviewer #2: Yes

Reviewer #2: The revisions have addressed the questions and concerns I had - congratulations to the authors on a very nice paper!

**Do you want your identity to be public for this peer review?** For information about this choice, including consent withdrawal, please see our Privacy Policy

Reviewer #2: No

---

## [Editor Report · Acceptance letter]

PONE-D-25-09385R1

PLOS ONE

Dear Dr. Ujino-Ihara,

I'm pleased to inform you that your manuscript has been deemed suitable for publication in PLOS ONE. Congratulations! Your manuscript is now being handed over to our production team.

Kind regards,

on behalf of

Dr. Shailender Kumar Verma

Academic Editor

PLOS ONE